# LMI-Based Results on Robust Exponential Passivity of Uncertain Neutral-Type Neural Networks with Mixed Interval Time-Varying Delays via the Reciprocally Convex Combination Technique

**Nayika Samorn [1], Narongsak Yotha [2], Pantiwa Srisilp [3] and Kanit Mukdasai [1,\*]**

1   Department of Mathematics, Faculty of Science, Khon Kaen University, Khon Kaen 40002, Thailand; nayika.s@kkumail.com
2   Department of Applied Mathematics and Statistics, Faculty of Science and Liberal Arts, Rajamangala University of Technology Isan, Nakhon Ratchasima 30000, Thailand; narongsak.yo@rmuti.ac.th
3   Rail System Institute, Rajamangala University of Technology Isan, Nakhon Ratchasima 30000, Thailand; pantiwa.sr@rmuti.ac.th
\*   Correspondence: kanit@kku.ac.th

**Abstract:** The issue of the robust exponential passivity analysis for uncertain neutral-type neural networks with mixed interval time-varying delays is discussed in this work. For our purpose, the lower bounds of the delays are allowed to be either positive or zero adopting the combination of the model transformation, various inequalities, the reciprocally convex combination, and suitable Lyapunov–Krasovskii functional. A new robust exponential passivity criterion is received and formulated in the form of linear matrix inequalities (LMIs). Moreover, a new exponential passivity criterion is also examined for systems without uncertainty. Four numerical examples indicate our potential results exceed the previous results.

**Keywords:** robust exponential passivity; neutral-type neural networks; Lyapunov–Krasovskii functional; interval time-varying delays

## 1. Introduction

Delayed dynamical systems have been proposed rather extensively because they can be exploited as models to illustrate the transportation systems, communication networks, teleportation systems, physical systems, and biological systems. Time delay regularly appeared in many practicals, and it is frequently a cause of instability and terrible performance. Largely, stability for systems with time delays in mainly divided into two categories; delay-independent and delay-dependent. Delay-independent stability criteria have a tendency to be more conservative, particularly for delays with small size; such criteria give no information on the size of the delay. On the other hand, delay-dependent stability criteria are connected with the size of the delay and generally provide a maximal delay size. Meanwhile, type of time delays are separated the into two types including information processing delay and information communication delay, which for which a lot of issues delayed scaled consensus have been presented [1]. Lately, practical engineering systems have examined the delays of which the lower bound of the delay is not limited to zero, which is called an interval time-varying delay. Several delay-interval-dependent criteria of systems are shown in [2–8]. Moreover, the neutral time delay is a type of delay which is currently drawing attention. This comes from the fact that the delay exists in systems both in its derivatives and state variables [9,10] which can be noticed in various fields such as mechanics, automatic control, distributed networks, heat exchanges, and robots in contact with rigid environments [11,12] etc.

Nowadays, neural networks are popularly discussed because they can be applied in many fields. Especially engineering and applied science including signal processing,

pattern recognition, industrial automation, image processing, parallel computation, industrial automation [13–18]) etc. Therefore, many researchers have been interested in studying neural networks with time-delays [5,19–22]. Furthermore, in neural networks, it might occur that there are connections between past state derivatives in the systems. As a result, it is more natural to consider neural networks with activation functions of past state derivative networks. Neural networks of this model are called neutral-type neural networks (NTNNs), which have appeared to be useful systems in a variety of applications, including population ecology, propagation, diffusion models and so on. Meanwhile, as is well known, in the biochemistry experiments of neural network dynamics, neural information may transfer across chemical reactivity, which results in a neutral-type process. Recently, the research on the above systems has been extensive and there are many findings in this area [8,23–26].

As the exponential stability is also important for testing stability because it can identify the rate of convergence of the system, which correlates with equilibrium points, accordingly, the exponential stability of various systems has also received a lot of attention from the researchers (for examples, see [19,21–23,25,26]). Meanwhile, the passivity theory is considered and has played an important role in the astonishing stability of time-delay systems [27,28]. It generally has its practical use in signal processing [29], complexity [30], chaos and synchronization control [31], and fuzzy control [32]. The main idea of the theory informs that the stability of the system can be perfectly maintained by the passivity's properties. Because of this, it can lead to general conclusions on the stability using only input—output characteristics. As a consequence, there are quite a few researchers who have been studying this issue. (e.g., [8,20–22,25,28–32]).The properties of exponential passivity for dynamical systems were studied in [33–35], who remark that exponential passivity implies passivity, but the opposite does not necessarily hold. Note that a lot of previous studies mainly attracted on work on stability and passivity analysis of neutral-type neural networks [8,23–26], and exponential passivity analysis of neural networks [21,22]. As far as we can tell, the robust exponential passivity of the uncertain neutral-type neural networks with mixed interval time-varying delays has never been presented.

In this paper, the issue of the new robust exponential passivity criterion is designed for uncertain NTNNs with mixed interval time-varying delays including discrete, neutral, and distributed delays. One of the aims of the criterion is to obtain the maximum upper bounds of time delays or maximum values of the rate of convergence. So, we concentrate on interval time-varying delays, in which the lower bounds are allowed to be either positive or zero. The model transformation, the various inequalities, and the reciprocally convex combination are adopted along with a suitable Lyapunov—Krasovskii functional when estimating their derivatives to improve the performance of the uncertain NTNNs. A new robust exponential passivity criterion is received and formulated in the form of LMIs. Moreover, a new exponential passivity criterion for NTNNs without uncertainty is also examined. The main contributions of this work are highlighted as follows: (i) the criterion proposed different from the NTNNs reported in [8,23–26]; (ii) the method suggested here can be used for the general neural networks with implied distributed time-varying delays [22] and the implied general neural networks implies [21]. Finally, we present some results that show the potential results exceed the results that previously seen.

**Notations**. $\mathbb{R}^n$ and $\mathbb{R}^{n \times r}$ denotes the n-dimensional Euclidean space and the set of all $n \times r$ real matrices, respectively. $B > 0$ ($B \geq 0$) means that the symmetric matrix $B$ is positive (semi-positive) definite; $B < 0$ ($B \leq 0$) means that the symmetric matrix $B$ is negative (semi-negative) definite. $I$ is the identity matrix with appropriate dimensions. $*$ represents the elements below the main diagonal of a symmetric matrix. $\dot{z}(t)$ denotes the upper right-hand derivative of $z$ at $t$. $z_t = \{z(t + \theta) : \theta \in [-\max\{\delta_2, \tau_2, \eta_2\}, 0]\}$. $\dot{V}(t, \phi) = \lim_{\theta \to 0^+} \sup\{t + \theta, z_{t+\theta}(t, \phi) - V(t, \phi)\}/\theta$ where $\phi(t)$ is the initial function that is continuously differentiable on $C([-\max\{\delta_2, \tau_2, \eta_2\}, 0], \mathbb{R}^n)$.

## 2. Preliminaries

First, we suggest the uncertain NTNNs, which are the form

$$
\begin{aligned}
\dot{z}(t) &= -(A + \Delta A(t))z(t) + (W_0 + \Delta W_0(t))g(z(t)) \\
&\quad + (W_1 + \Delta W_1(t))g(z(t - \delta(t))) + (W_2 + \Delta W_2(t))\dot{z}(t - \tau(t)) \\
&\quad + (W_3 + \Delta W_3(t))\int_{t-\eta(t)}^{t} g(z(s))ds + u(t), \quad t \geq 0, \\
y(t) &= C_0 g(z(t)) + C_1 g(z(t - \delta(t))) + C_2 \int_{t-\eta(t)}^{t} g(z(s))ds + C_3 u(t), \\
z(t) &= \phi(t), \quad t \in [-\max\{\tau_2, \delta_2, \eta_2\}, 0],
\end{aligned}
\tag{1}
$$

where $z(t) = [z_1(t), z_2(t), ..., z_n(t)] \in \mathbb{R}^n$ is the neuron state vector, $y(t)$ is the output vector of neuron networks, $A = [a_i]$ is a diagonal matrix with $a_i > 0, i = 1, 2, ..., n$, $W_0$ is the connection weight matrix, $W_1$, $W_2$ and $W_3$ are the delayed connection weight matrices. $C_0$, $C_1$, $C_2$, and $C_3$, are given real matrices, $u(t) \in \mathbb{R}^n$ is an external input vector to neurons, the continuous functions $\phi(t)$ and $\varphi(t)$ are the initial conditions.

The delays $\tau(t)$, $\delta(t)$ and $\eta(t)$ satisfy

$$
0 \leq \tau_1 \leq \tau(t) \leq \tau_2, \quad \dot{\tau}(t) \leq \tau_d,
\tag{2}
$$

$$
0 \leq \delta_1 \leq \delta(t) \leq \delta_2, \quad \dot{\delta}(t) \leq \delta_d,
\tag{3}
$$

$$
0 \leq \eta_1 \leq \eta(t) \leq \eta_2,
\tag{4}
$$

where $\tau_1$, $\tau_2$, $\delta_1$, $\delta_2$, $\eta_1$, $\eta_2$, $\tau_d$ and $\delta_d$ are non-negative real constants.

**Assumption 1.** *The activation function* $g(z(t)) = [g_1(z_1(t)), g_2(z_2(t)), ..., g_n(z_n(t))]^T \in \mathbb{R}^n$ *is assumed to satisfy the following condition*

$$
0 \leq \frac{g_i(\zeta_1) - g_i(\zeta_2)}{\zeta_1 - \zeta_2} \leq l_i, \qquad g(0) = 0, \qquad \zeta_1, \zeta_2 \in \mathbb{R}, \qquad \zeta_1 \neq \zeta_2, i = 1, 2, ..., n,
\tag{5}
$$

*where* $l_i, i = 1, 2, ..., n$ *are positive real constants, we denote* $L = [l_i]$, $i = 1, 2, ..., n$ *as a diagonal matrix.*

The uncertainty matrices. $\Delta A(t)$, $\Delta W_0(t)$, $\Delta W_1(t)$, $\Delta W_2(t)$, and $\Delta W_3(t)$, are assumed to be of the form

$$
\begin{bmatrix} \Delta A(t) & W_0(t) & \Delta W_1(t) & \Delta W_2(t) & \Delta W_3 \end{bmatrix} = E\Delta(t)\begin{bmatrix} G_a & G_0 & G_1 & G_2 & G_3 \end{bmatrix},
$$

where $E$, $G_a$, and $G_i$, $i = 0, 1, 2, 3$, are known real constant matrices; the uncertainty matrix $\Delta(t)$ satisfies

$$
\Delta(t) = F(t)[I - JF(t)]^{-1},
\tag{6}
$$

is said to be admissible where $J$ is an unknown matrix satisfying

$$
I - JJ^T > 0,
\tag{7}
$$

and the uncertainty matrix $F(t)$ is satisfying

$$
F(t)^T F(t) \geq 0.
\tag{8}
$$

**Assumption 2.** *All eigenvalues of the matrix* $W_2 + \Delta W_2(t)$ *are inside the unit circle.*

Then, the following Definition and Lemmas are methods that are use to prove our main results.

**Definition 1** ([25]). *The system* (1) *is said to be robust and exponentially passive from input $u(t)$ to output $y(t)$, if there exists an exponential Lyapunov function $V(z_t)$, and a constant $\rho > 0$ such that for all $u(t)$, all initial conditions $z(t_0)$, all $t \geq t_0$, the following inequality holds:*

$$\dot{V}(z_t) + \rho V(z_t) \leq 2z^T(t)u(t); \quad t \geq t_0,$$

*where $\dot{V}(z_t)$ denotes the total derivative of $V(z_t)$ along the state trajectories $z(t)$ of the system* (1)

**Lemma 1** ((*Jensen's inequality*) [14]). *Let $Q \in \mathbb{R}^{n \times n}$, $Q = Q^T > 0$ be any constant matrix, $\delta_2$ be positive real constant and $\omega : [-\delta_2, 0] \to \mathbb{R}^n$ be vector-valued function. Then,*

$$-\delta_2 \int_{t-\delta_2}^t \omega^T(s)Q\omega(s)ds \leq -\Big(\int_{t-\delta_2}^t \omega(s)ds\Big)^T Q\Big(\int_{t-\delta_2}^t \omega(s)ds\Big).$$

**Lemma 2** ([36]). *Let $f_1, f_2, \ldots, f_N : \mathbb{R}^n \to \mathbb{R}$ have positive values in an open subset $D$ of $\mathbb{R}^n$. Then, the reciprocally convex combination of $f_i$ over $D$ satisfies*

$$\min_{\{\alpha_i | \alpha_i > 0, \sum_i \alpha_i = 1\}} \sum_i \frac{1}{\alpha_i} f_i(t) = \sum_i f_i(t) + \max_{g_{i,j}(t)} \sum_{i \neq j} g_{i,j}(t),$$

*subject to*

$$\left\{ g_{i,j} : \mathbb{R}^n \to \mathbb{R}, g_{i,j} = g_{j,i}, \begin{bmatrix} f_i(t) & g_{i,j}(t) \\ g_{j,i}(t) & f_j(t) \end{bmatrix} \geq 0 \right\}.$$

**Lemma 3** ([6]). *For $Q \in \mathbb{R}^{n \times n}$, $Q = Q^T > 0$, and any continuously differentiable function $z : [\sigma_1, \sigma_2] \to \mathbb{R}^n$, the following inequality holds:*

$$
\begin{aligned}
(\sigma_2 - \sigma_1) \int_{\sigma_1}^{\sigma_2} \dot{z}^T(s)Q\dot{z}(s)ds &\geq \Omega_1^T Q\Omega_1 + 3\Omega_2^T Q\Omega_2 + 5\Omega_3^T Q\Omega_3 + 7\Omega_4^T Q\Omega_4, \\
\int_{\sigma_1}^{\sigma_2} \int_{\theta}^{\sigma_2} \dot{z}^T(s)Q\dot{z}(s)dsd\theta &\geq 2\Omega_5^T Q\Omega_5 + 4\Omega_6^T Q\Omega_6 + 6\Omega_7^T Q\Omega_7, \\
\int_{\sigma_1}^{\sigma_2} \int_{\sigma_1}^{\theta} \dot{z}^T(s)Q\dot{z}(s)dsd\theta &\geq 2\Omega_8^T Q\Omega_8 + 4\Omega_9^T Q\Omega_9 + 6\Omega_{10}^T Q\Omega_{10},
\end{aligned}
$$

*where*

$$\Omega_1 = z(\sigma_2) - z(\sigma_1), \quad \Omega_2 = z(\sigma_2) + z(\sigma_1) - \frac{2}{\sigma_2 - \sigma_1} \int_{\sigma_1}^{\sigma_2} z(s)ds,$$

$$\Omega_3 = z(\sigma_2) - z(\sigma_1) + \frac{6}{\sigma_2 - \sigma_1} \int_{\sigma_1}^{\sigma_2} z(s)ds - \frac{12}{(\sigma_2 - \sigma_1)^2} \int_{\sigma_1}^{\sigma_2} \int_{\theta}^{\sigma_2} z(s)dsd\theta,$$

$$\Omega_4 = z(\sigma_2) + z(\sigma_1) - \frac{12}{\sigma_2 - \sigma_1} \int_{\sigma_1}^{\sigma_2} z(s)ds + \frac{60}{(\sigma_2 - \sigma_1)^2} \int_{\sigma_1}^{\sigma_2} \int_{\theta}^{\sigma_2} z(s)dsd\theta$$

$$-\frac{120}{(\sigma_2 - \sigma_1)^3} \int_{\sigma_1}^{\sigma_2} \int_u^{\sigma_2} \int_{\theta}^{\sigma_2} z(s)dsd\theta du, \quad \Omega_5 = z(\sigma_2) - \frac{1}{\sigma_2 - \sigma_1} \int_{\sigma_1}^{\sigma_2} z(s)ds,$$

$$\Omega_6 = z(\sigma_2) + \frac{2}{\sigma_2 - \sigma_1} \int_{\sigma_1}^{\sigma_2} z(s)ds - \frac{6}{(\sigma_2 - \sigma_1)^2} \int_{\sigma_1}^{\sigma_2} \int_{\theta}^{\sigma_2} z(s)dsd\theta,$$

$$\Omega_7 = z(\sigma_2) - \frac{3}{\sigma_2 - \sigma_1} \int_{\sigma_1}^{\sigma_2} z(s)ds + \frac{24}{(\sigma_2 - \sigma_1)^2} \int_{\sigma_1}^{\sigma_2} \int_{\theta}^{\sigma_2} z(s)dsd\theta$$

$$-\frac{60}{(\sigma_2 - \sigma_1)^3} \int_{\sigma_1}^{\sigma_2} \int_u^{\sigma_2} \int_{\theta}^{\sigma_2} z(s)dsd\theta du, \quad \Omega_8 = z(\sigma_1) - \frac{1}{\sigma_2 - \sigma_1} \int_{\sigma_1}^{\sigma_2} z(s)ds,$$

$$\Omega_9 = z(\sigma_1) - \frac{4}{\sigma_2 - \sigma_1} \int_{\sigma_1}^{\sigma_2} z(s)ds + \frac{6}{(\sigma_2 - \sigma_1)^2} \int_{\sigma_1}^{\sigma_2} \int_{\theta}^{\sigma_2} z(s)dsd\theta,$$

$$\Omega_{10} = z(\sigma_1) - \frac{9}{\sigma_2 - \sigma_1} \int_{\sigma_1}^{\sigma_2} z(s)ds + \frac{36}{(\sigma_2 - \sigma_1)^2} \int_{\sigma_1}^{\sigma_2} \int_{\theta}^{\sigma_2} z(s)dsd\theta$$
$$- \frac{60}{(\sigma_2 - \sigma_1)^3} \int_{\sigma_1}^{\sigma_2} \int_u^{\sigma_2} \int_{\theta}^{\sigma_2} z(s)dsd\theta du.$$

**Lemma 4** ([37]). *For any real constant matrices of appropriate dimensions M, S and N with $M = M^T$, and $\Delta(t)$ is given as constant by (6)–(8), then*

$$M + S\Delta(t)N + N^T\Delta(t)^T S^T < 0,$$

*holds if and only if*

$$\begin{bmatrix} M & S & \beta N^T \\ * & -\beta I & \beta J^T \\ * & * & -\beta I \end{bmatrix} < 0,$$

*where $\beta$ is any positive real constant.*

### 3. Main Results

This section intended to develop new criteria of system (1) with conditions (2)–(4). We separate the consideration into two parts. In the first part, we consider the nominal system, then suggest our main system, in which new criteria of systems are introduced via LMIs approach.

**Theorem 1.** *Assume that Assumptions 1 and 2 hold. For given scalars $\delta_1$, $\delta_2$, $\delta_d$, $\tau_1$, $\tau_u$, $\tau_d$, $\eta_1$, $\eta_2$, $\rho$ with conditions (2)–(4) and $\rho > 0$, if there exist matrices $P > 0$, $M_n$, $n = 1, 2, \cdots, 6$, $\begin{bmatrix} M_1 & M_2 \\ * & M_3 \end{bmatrix} > 0$, $\begin{bmatrix} M_4 & M_5 \\ * & M_6 \end{bmatrix} > 0$, $N_j > 0$, $O_j > 0$, $S_j > 0$, $T_j > 0$, $j = 1, 2, 3$, any diagonal matrices $D_1 > 0$, $D_2 > 0$, $U_i > 0$, $i = 1, 2, 3, 4$, any appropriate dimensional matrices $X_k$, $Y_l$, $Z_i$, $k = 1, 2, \cdots, 7, l = 1, 2, \cdots, 5, i = 1, 2, 3, 4$, satisfying the following*

$$\Lambda < 0, \tag{9}$$

$$\begin{bmatrix} O_1 + O_2 & Z_i \\ * & O_1 + O_3 \end{bmatrix} \geq 0, \tag{10}$$

*then the system (11) is exponential passive, where $\Lambda$ is defined in Appendix A.*

**Proof.** Firstly, we propose the exponential passivity analysis for the nominal system

$$\dot{z}(t) = -Az(t) + W_0g(z(t)) + W_1g(z(t - \delta(t))) + W_2\dot{z}(t - \tau(t)) + W_3 \int_{t-\eta(t)}^t g(z(t)) + u(t), \quad t \geq 0. \tag{11}$$

Second, modify the system (11) in terms of model transformation, which is the form as follows

$$\dot{z}(t) = w(t), \tag{12}$$
$$0 = -w(t) - Az(t) + W_0g(z(t)) + W_1g(z(t - \delta(t))) + W_2\dot{z}(t - \tau(t))$$
$$+ W_3 \int_{t-\eta(t)}^t g(z(t)) + u(t). \tag{13}$$

Then, the Lyapunov–Krasovskii functional is designed for the system (11) and (13):

$$V(z_t) = \sum_{i=1}^6 V_i(z_t),$$

where

$$V_1(z_t) = z^T(t)Pz(t) + 2\sum_{i=1}^{n}\int_0^{z_i(t)}\left[d_{1i}(g_i(s)) + d_{2i}(l_i s - g_i(s))\right]ds,$$

$$V_2(z_t) = \int_{t-\delta_2}^{t-\delta_1} e^{2\alpha(s-t)}\begin{bmatrix} z(s) \\ g(z(s)) \end{bmatrix}^T\begin{bmatrix} M_1 & M_2 \\ * & M_3 \end{bmatrix}\begin{bmatrix} z(s) \\ g(z(s)) \end{bmatrix}ds$$

$$+ \int_{t-\delta(t)}^{t} e^{2\alpha(s-t)}\begin{bmatrix} z(s) \\ g(z(s)) \end{bmatrix}^T\begin{bmatrix} M_4 & M_5 \\ * & M_6 \end{bmatrix}\begin{bmatrix} z(s) \\ g(z(s)) \end{bmatrix}ds,$$

$$V_3(z_t) = \delta_1\int_{t-\delta_1}^{t}\int_\theta^{t} e^{2\alpha(s-t)}\dot{z}^T(s)N_1\dot{z}(s)dsd\theta$$

$$+ \int_{t-\delta_1}^{t}\int_u^{t}\int_\theta^{t} e^{2\alpha(s-t)}\dot{z}^T(s)N_2\dot{z}(s)dsd\theta du$$

$$+ \int_{t-\delta_1}^{t}\int_{t-\delta_1}^{u}\int_\theta^{t} e^{2\alpha(s-t)}\dot{z}^T(s)N_3\dot{z}(s)dsd\theta du,$$

$$V_4(z_t) = (\delta_2 - \delta_1)\int_{t-\delta_2}^{t-\delta_1}\int_\theta^{t} e^{2\alpha(s-t)}\dot{z}^T(s)O_1\dot{z}(s)dsd\theta$$

$$+ \int_{t-\delta_2}^{t-\delta_1}\int_u^{t-\delta_1}\int_\theta^{t} e^{2\alpha(s-t)}z^T(s)O_2 z(s)dsd\theta du$$

$$+ \int_{t-\delta_2}^{t-\delta_1}\int_{t-\delta_2}^{u}\int_\theta^{t} e^{2\alpha(s-t)}\dot{z}^T O_3\dot{z}(s)dsd\theta du,$$

$$V_5(z_t) = \int_{t-\tau_2}^{t} e^{2\alpha(s-t)}\dot{z}^T(s)S_1\dot{z}(s)ds + \int_{t-\tau(t)}^{t} e^{2\alpha(s-t)}\dot{z}^T(s)S_2\dot{z}(s)ds$$

$$+ \int_{t-\tau_1}^{t} e^{2\alpha(s-t)}\dot{z}^T(s)S_3\dot{z}(s)ds,$$

$$V_6(z_t) = \eta_2\int_{t-\eta_2}^{t}\int_\theta^{t} e^{2\alpha(s-t)}g^T(z(s))(T_1 + T_2)g(z(s))dsd\theta$$

$$+ \eta_1\int_{t-\eta_1}^{t}\int_\theta^{t} e^{2\alpha(s-t)}g^T(z(s))T_3 g(z(s))dsd\theta.$$

From the time derivative of $V_1(z_t)$ along the trajectory of system (11) and (13), we obtain

$$\dot{V}_1(z_t) = 2\begin{bmatrix} z(t) \\ \int_{t-\delta_2}^{t}\dot{z}(s)ds \\ g(z(t)) \\ g(z(t-\delta(t))) \end{bmatrix}^T\begin{bmatrix} P & Q_1^T & Q_5^T & Q_9^T \\ 0 & Q_2^T & Q_6^T & Q_{10}^T \\ 0 & Q_3^T & Q_7^T & Q_{11}^T \\ 0 & Q_4^T & Q_8^T & Q_{12}^T \end{bmatrix}\begin{bmatrix} \dot{z}(t) \\ 0 \\ 0 \\ 0 \end{bmatrix}$$

$$+ 2g^T(z(t))D_1\dot{z}(t) + 2z^T(t)D_2 L\dot{z}(t) - 2g^T(z(t))D_2\dot{z}(t)$$

$$= 2z^T(t)P\left[ -Az(t) + W_0 g(z(t)) + W_1 g(z(t-\delta(t))) + W_2\dot{z}(t-\tau(t))\right.$$

$$+ W_3\int_{t-\eta(t)}^{t} g(z(s))ds + u(t)\bigg] + \left[z^T(t)Q_1^T + \int_{t-\delta_2}^{t}\dot{z}^T(s)dsQ_2^T\right.$$

$$+ g^T(z(t))(s)Q_3^T + g^T(z(t-\delta(t))Q_4^T\bigg]\left[z(t) - z(t-\delta(t))\right.$$

$$- \int_{t-\delta_2}^{t}\dot{z}(s)ds\bigg] + \left[z^T(t)Q_5^T + \int_{t-\delta_2}^{t}\dot{z}^T(s)dsQ_6^T + g^T(z(t))(s)Q_7^T\right.$$

$$+ g^T(z(t-\delta(t))Q_8^T\bigg]\left[z(t) - z(t-\delta(t)) - \int_{t-\delta_2}^{t}\dot{z}(s)ds\right] + \left[z^T(t)Q_9^T\right.$$

$$+ \int_{t-\delta_2}^{t} \dot{z}^T(s)ds Q_{10}^T + g^T(z(t))(s)Q_{11}^T + g^T(z(t - \delta(t))Q_{12}^T \Big] \Big[ -w(t)$$

$$-Az(t) + W_0 g(z(t)) + W_1 g(z(t - \delta(t))) + W_2 \dot{z}(t - \tau(t))$$

$$+W_3 \int_{t-\eta(t)}^{t} g(z(t)) + u(t) \Big] + 2g^T(z(t))D_1\dot{z}(t) + 2z^T(t)D_2L\dot{z}(t)$$

$$-2g^T(z(t))D_2\dot{z}(t) + 2\alpha z^T(t)P_1 z(t) + 4\alpha g^T(z(t))D_1 z(t)$$

$$+4\alpha \Big[ z^T(t)L - g^T(z(t)) \Big] D_2 z(t) - 2\alpha V_1(z_t).$$

Calculating $\dot{V}_2(z_t)$ leads to

$$\dot{V}_2(z_t) \leq e^{-2\alpha\delta_1} \begin{bmatrix} z(t - \delta_1) \\ g(z(t - \delta_1)) \end{bmatrix}^T \begin{bmatrix} M_1 & M_2 \\ * & M_3 \end{bmatrix} \begin{bmatrix} z(t - \delta_1) \\ g(z(t - \delta_1)) \end{bmatrix}$$

$$-e^{-2\alpha\delta_2} \begin{bmatrix} z(t - \delta_2) \\ g(z(t - \delta_2)) \end{bmatrix}^T \begin{bmatrix} M_1 & M_2 \\ * & M_3 \end{bmatrix} \begin{bmatrix} z(t - \delta_2) \\ g(z(t - \delta_2)) \end{bmatrix}$$

$$+ \begin{bmatrix} z(t) \\ g(z(t)) \end{bmatrix}^T \begin{bmatrix} M_4 & M_5 \\ * & M_6 \end{bmatrix} \begin{bmatrix} z(t) \\ g(z(t)) \end{bmatrix} + (\delta_d - e^{-2\alpha\delta_2}) \begin{bmatrix} z(t - \delta(t)) \\ g(z(t - \delta(t))) \end{bmatrix}^T$$

$$\times \begin{bmatrix} M_4 & M_5 \\ * & M_6 \end{bmatrix} \begin{bmatrix} z(t - \delta(t)) \\ g(z(t - \delta(t))) \end{bmatrix} - 2\alpha V_2(z_t).$$

By employing Lemma 3 to estimate the integral terms in $\dot{V}_3(z_t)$, we readily obtain

$$\dot{V}_3(z_t) \leq \dot{z}(t) \Big[ \delta_1 N_1 + \frac{\delta_1^2}{2}(N_2 + N_3) \Big] \dot{z}(t) - e^{-2\alpha\delta_2} \Big\{ \Omega_{1\,[t-\delta_1],t}^T N_1 \Omega_{1\,[t-\delta_1,t]}$$

$$+3\Omega_{2\,[t-\delta_1,t]}^T N_1 \Omega_{2[t-\delta_1,t]} + 5\Omega_{3\,[t-\delta_1,t]}^T N_1 \Omega_{3[t-\delta_1,t]}^T + 7\Omega_{4\,[t-\delta_1,t]}^T N_1$$

$$\times \Omega_{4[t-\delta_1,t]} + 2\Omega_{5\,[t-\delta_1,t]}^T N_2 \Omega_{5[t-\delta_1,t]} + 4\Omega_{6\,[t-\delta_1,t]}^T N_2 \Omega_{6[t-\delta_1,t]}$$

$$+6\Omega_{7\,[t-\delta_1,t]}^T N_2 \Omega_{7[t-\delta_1,t]}^T + 2\Omega_{8\,[t-\delta_1,t]}^T N_3 \Omega_{8[t-\delta_1,t]} + 4\Omega_{9\,[t-\delta_1,t]}^T N_3$$

$$\times \Omega_{9[t-\delta_1,t]} + 6\Omega_{10[t-\delta_1,t]}^T N_3 \Omega_{10[t-\delta_1,t]} \Big\} - 2\alpha V_3(z_t),$$

where

$$\Omega_1[\sigma_1, \sigma_2] = z(\sigma_2) - z(\sigma_1), \ \Omega_2[\sigma_1, \sigma_2] = z(\sigma_2) + z(\sigma_1) - \frac{2}{\sigma_2 - \sigma_1} \int_{\sigma_1}^{\sigma_2} z(s)ds,$$

$$\Omega_3[\sigma_1, \sigma_2] = z(\sigma_2) - z(\sigma_1) + \frac{6}{\sigma_2 - \sigma_1} \int_{\sigma_1}^{\sigma_2} z(s)ds - \frac{12}{(\sigma_2 - \sigma_1)^2} \int_{\sigma_1}^{\sigma_2} \int_{\theta}^{\sigma_2} z(s)ds d\theta,$$

$$\Omega_4[\sigma_1, \sigma_2] = z(\sigma_2) + z(\sigma_1) - \frac{12}{\sigma_2 - \sigma_1} \int_{\sigma_1}^{\sigma_2} z(s)ds + \frac{60}{(\sigma_2 - \sigma_1)^2} \int_{\sigma_1}^{\sigma_2} \int_{\theta}^{\sigma_2} z(s)ds d\theta$$

$$-\frac{120}{(\sigma_2 - \sigma_1)^3} \int_{\sigma_1}^{\sigma_2} \int_{u}^{\sigma_2} \int_{\theta}^{\sigma_2} z(s)ds d\theta du, \ \Omega_5[\sigma_1, \sigma_2] = z(\sigma_2) - \frac{1}{\sigma_2 - \sigma_1} \int_{\sigma_1}^{\sigma_2} z(s)ds,$$

$$\Omega_6[\sigma_1, \sigma_2] = z(\sigma_2) + \frac{2}{\sigma_2 - \sigma_1} \int_{\sigma_1}^{\sigma_2} z(s)ds - \frac{6}{(\sigma_2 - \sigma_1)^2} \int_{\sigma_1}^{\sigma_2} \int_{\theta}^{\sigma_2} z(s)ds d\theta,$$

$$\Omega_7[\sigma_1, \sigma_2] = z(\sigma_2) - \frac{3}{\sigma_2 - \sigma_1} \int_{\sigma_1}^{\sigma_2} z(s)ds + \frac{24}{(\sigma_2 - \sigma_1)^2} \int_{\sigma_1}^{\sigma_2} \int_{\theta}^{\sigma_2} z(s)ds d\theta$$

$$-\frac{60}{(\sigma_2 - \sigma_1)^3} \int_{\sigma_1}^{\sigma_2} \int_{u}^{\sigma_2} \int_{\theta}^{\sigma_2} z(s)ds d\theta du, \ \Omega_8[\sigma_1, \sigma_2] = z(\sigma_1) - \frac{1}{\sigma_2 - \sigma_1} \int_{\sigma_1}^{\sigma_2} z(s)ds,$$

$$\Omega_9[\sigma_1, \sigma_2] = z(\sigma_1) - \frac{4}{\sigma_2 - \sigma_1} \int_{\sigma_1}^{\sigma_2} z(s)ds + \frac{6}{(\sigma_2 - \sigma_1)^2} \int_{\sigma_1}^{\sigma_2} \int_{\theta}^{\sigma_2} z(s)ds d\theta,$$

$$\Omega_{10}[\sigma_1, \sigma_2] = z(\sigma_1) - \frac{9}{\sigma_2 - \sigma_1} \int_{\sigma_1}^{\sigma_2} z(s)ds + \frac{36}{(\sigma_2 - \sigma_1)^2} \int_{\sigma_1}^{\sigma_2} \int_{\theta}^{\sigma_2} z(s)dsd\theta$$

$$-\frac{60}{(\sigma_2 - \sigma_1)^3} \int_{\sigma_1}^{\sigma_2} \int_{u}^{\sigma_2} \int_{\theta}^{\sigma_2} z(s)dsd\theta du, \ t - \delta_2 \leq \sigma_1 \leq \sigma_2 \leq t.$$

By calculating the derivative of $V_4(z_t)$, we obtain

$$
\begin{aligned}
\dot{V}_4(z_t) \leq{} & \dot{z}(t)\left[(\delta_2 - \delta_1)^2 O_1 + \frac{(\delta_2 - \delta_1)^2}{2}(O_2 + O_3)\right]\dot{z}(t) - e^{-2\alpha\delta_2}(\delta_2 - \delta_1) \\
& \times \int_{t-\delta_2}^{t-\delta_1} \dot{z}^T(s)O_2\dot{z}(s)ds - e^{-2\alpha\delta_2} \int_{t-\delta_2}^{t-\delta_1} \int_{\theta}^{t-\delta_1} \dot{z}^T(s)O_2\dot{z}(s)dsd\theta \\
& - e^{-2\alpha\delta_2} \int_{t-\delta_2}^{t-\delta_1} \int_{t-\delta_2}^{\theta} \dot{z}^T(s)O_3\dot{z}(s)dsd\theta - 2\alpha V_4(z_t) \\
\leq{} & \dot{z}(t)\left[(\delta_2 - \delta_1)^2 O_1 + \frac{(\delta_2 - \delta_1)^2}{2}(O_2 + O_3)\right]\dot{z}(t) - e^{-2\alpha\delta_2}\Bigg\{(\delta_2 - \delta_1) \\
& \times \int_{t-\delta_2}^{t-\delta(t)} \dot{z}^T(s)O_1\dot{z}(s)ds + (\delta_2 - \delta_1) \int_{t-\delta(t)}^{t-\delta_1} \dot{z}^T(s)O_1\dot{z}(s)ds \\
& + \int_{t-\delta_2}^{t-\delta(t)} \int_{\theta}^{t-\delta(t)} \dot{z}^T(s)O_2\dot{z}(s)ds + \int_{t-\delta(t)}^{t-\delta_1} \int_{\theta}^{t-\delta_1} \dot{z}^T(s)O_2\dot{z}(s)ds \\
& + (\delta_2 - \delta(t)) \int_{t-\delta(t)}^{t-\delta_1} \dot{z}^T(s)O_2\dot{z}(s)ds + \int_{t-\delta_2}^{t-\delta(t)} \int_{t-\delta_2}^{\theta} \dot{z}^T(s)O_3\dot{z}(s)ds \\
& + \int_{t-\delta(t)}^{t-\delta_1} \int_{t-\delta(t)}^{\theta} \dot{z}^T(s)O_3\dot{z}(s)ds + (\delta(t) - \delta_1) \int_{t-\delta_2}^{t-\delta(t)} \dot{z}^T(s)O_3\dot{z}(s)ds\Bigg\} \\
& - 2\alpha V_4(z_t).
\end{aligned}
$$

Since $O_1 > 0, O_2 > 0$ and $O_3 > 0$, using Lemma 3, we observe that

$$
\begin{aligned}
& -e^{-2\alpha\delta_2}\Bigg\{(\delta_2 - \delta_1)\int_{t-\delta(t)}^{t-\delta_1} \dot{z}^T(s)O_1\dot{z}(s)ds + (\delta_2 - \delta_1)\int_{t-\delta_2}^{t-\delta(t)} \dot{z}^T(s)O_1\dot{z}(s)ds \\
& + (\delta_2 - \delta(t))\int_{t-\delta(t)}^{t-\delta_1} \dot{z}^T(s)O_2\dot{z}(s)ds + (\delta(t) - \delta_1)\int_{t-\delta_2}^{t-\delta(t)} \dot{z}^T(s)O_3\dot{z}(s)ds\Bigg\} \\
\leq{} & e^{-2\alpha\delta_2}\Bigg\{-\frac{(\delta_2 - \delta_1)}{(\delta(t) - \delta_1)}\Big(\Omega_1^T{}_{[t-\delta(t),t-\delta_1]}(O_1 + O_2)\Omega_{1[t-\delta(t),t-\delta_1]} \\
& + 3\Omega_2^T{}_{[t-\delta(t),t-\delta_1]}(O_1 + O_2)\Omega_{2[t-\delta(t),t-\delta_1]} + 5\Omega_3^T{}_{[t-\delta(t),t-\delta_1]}(O_1 + O_2) \\
& \times \Omega_{3[t-\delta(t),t-\delta_1]} + 7\Omega_4^T{}_{[t-\delta(t),t-\delta_1]}(O_1 + O_2)\Omega_{4[t-\delta(t),t-\delta_1]}\Big) \\
& + \Big(\Omega_1^T{}_{[t-\delta(t),t-\delta_1]}O_2\Omega_1^T{}_{[t-\delta(t),t-\delta_1]} + 3\Omega_2^T{}_{[t-\delta(t),t-\delta_1]}O_2\Omega_{2[t-\delta(t),t-\delta_1]} \\
& + 5\Omega_2^T{}_{[t-\delta(t),t-\delta_1]}O_2\Omega_{2[t-\delta(t),t-\delta_1]} + 7\Omega_2^T{}_{[t-\delta(t),t-\delta_1]}O_2\Omega_{2[t-\delta(t),t-\delta_1]}\Big) \\
& - \frac{(\delta_2 - \delta_1)}{(\delta_2 - \delta(t))}\Big(\Omega_1^T{}_{[t-\delta_2,t-\delta(t))]}(O_1 + O_3)\Omega_{1[t-\delta_2,t-\delta(t))]} \\
& + 3\Omega_2^T{}_{[t-\delta_2,t-\delta(t))]}(O_1 + O_3)\Omega_{2[t-\delta_2,t-\delta(t))]} + 5\Omega_3^T{}_{[t-\delta_2,t-\delta(t))]}(O_1 \\
& + O_3)\Omega_{3[t-\delta_2,t-\delta(t))]} + 7\Omega_4^T{}_{[t-\delta_2,t-\delta(t))]}(O_1 + O_3)\Omega_{4[t-\delta_2,t-\delta(t))]}\Big)
\end{aligned}
$$

$$+\left(\Omega_{1\,[t-\delta_2,t-\delta(t))]}^T O_3\Omega_{1\,[t-\delta_2,t-\delta(t))]}^T + 3\Omega_{2\,[t-\delta_2,t-\delta(t))]}^T O_3\Omega_{2\,[t-\delta_2,t-\delta(t))]}\right.$$

$$\left.\left.+5\Omega_{3\,[t-\delta_2,t-\delta(t))]}^T O_3\Omega_{3[t-\delta_2,t-\delta(t))]} + 7\Omega_{4\,[t-\delta_2,t-\delta(t))]}^T O_3\Omega_{4[t-\delta_2,t-\delta(t))]}\right)\right\}.$$

Next, we use the reciprocally convex combination technique to estimate inequality following

$$-e^{-2\alpha\delta_2}\left\{(\delta_2-\delta_1)\int_{t-\delta(t)}^{t-\delta_1}\dot{z}^T(s)O_1\dot{z}(s)ds + (\delta_2-\delta_1)\int_{t-\delta_2}^{t-\delta(t)}\dot{z}^T(s)O_1\dot{z}(s)ds\right.$$

$$\left.+(\delta_2-\delta(t))\int_{t-\delta(t)}^{t-\delta_1}\dot{z}^T(s)O_2\dot{z}(s)ds + (\delta(t)-\delta_1)\int_{t-\delta_2}^{t-\delta(t)}\dot{z}^T(s)O_3\dot{z}(s)ds\right\}$$

$$\leq -e^{-2\alpha\delta_2}\left\{\Omega_{1\,[t-\delta(t),t-\delta_1]}^T O_1\Omega_{1[t-\delta(t),t-\delta_1]} + 3\Omega_{2\,[t-\delta(t),t-\delta_1]}^T O_1\Omega_{2[t-\delta(t),t-\delta_1]}\right.$$

$$+5\Omega_{3\,[t-\delta(t),t-\delta_1]}^T O_1\Omega_{3[t-\delta(t),t-\delta_1]} + 7\Omega_{4\,[t-\delta(t),t-\delta_1]}^T O_1\Omega_{4[t-\delta(t),t-\delta_1]}$$

$$+\Omega_{1\,[t-\delta_2,t-\delta(t)]}^T O_1\Omega_{1[t-\delta_2,t-\delta(t)]} + 3\Omega_{2\,[t-\delta_2,t-\delta(t)]}^T O_1\Omega_{2[t-\delta_2,t-\delta(t)]}$$

$$+5\Omega_{3\,[t-\delta_2,t-\delta(t)]}^T O_1\Omega_{3[t-\delta_2,t-\delta(t)]} + 7\Omega_{4\,[t-\delta_2,t-\delta(t)]}^T O_1\Omega_{4[t-\delta_2,t-\delta(t)]}$$

$$+\Omega_{1\,[t-\delta(t),t-\delta_1]}^T Z_1\Omega_{1[t-\delta_2,t-\delta(t)]} + \Omega_{1\,[t-\delta_2,t-\delta(t)]}^T Z_1^T\Omega_{1[t-\delta(t),t-\delta_1]}$$

$$+3\Omega_{2\,[t-\delta(t),t-\delta_1]}^T Z_2\Omega_{2[t-\delta_2,t-\delta(t)]} + 3\Omega_{2\,[t-\delta_2,t-\delta(t)]}^T Z_2^T\Omega_{2[t-\delta(t),t-\delta_1]}$$

$$+5\Omega_{3\,[t-\delta(t),t-\delta_1]}^T Z_3\Omega_{3[t-\delta_2,t-\delta(t)]} + 5\Omega_{3\,[t-\delta_2,t-\delta(t)]}^T Z_3^T\Omega_{3[t-\delta(t),t-\delta_1]}$$

$$\left.+7\Omega_{4\,[t-\delta(t),t-\delta_1]}^T Z_4\Omega_{4[t-\delta_2,t-\delta(t)]} + 7\Omega_{4\,[t-\delta_2,t-\delta(t)]}^T Z_4^T\Omega_{4[t-\delta(t),t-\delta_1]}\right\}.$$

Then, we estimate $\dot{V}_4(z_t)$ as

$$\dot{V}_4(z_t) \leq \dot{z}(t)\left[(\delta_2-\delta_1)^2 O_1 + \frac{(\delta_2-\delta_1)^2}{2}(O_2+O_3)\right]\dot{z}(t)$$

$$-e^{-2\alpha\delta_2}\left\{2\Omega_{5\,[t-\delta(t),t-\delta_1]}^T O_2\Omega_{5[t-\delta(t),t-\delta_1]} + 4\Omega_{6\,[t-\delta(t),t-\delta_1]}^T O_2\right.$$

$$\times\Omega_{6[t-\delta(t),t-\delta_1]} + 6\Omega_{7\,[t-\delta(t),t-\delta_1]}^T O_2\Omega_{7\,[t-\delta(t),t-\delta_1]}^T + 2\Omega_{5\,[t-\delta_2,t-\delta(t)]}^T$$

$$\times O_2\Omega_{5[t-\delta_2,t-\delta(t)]} + 4\Omega_{6\,[t-\delta_2,t-\delta(t)]}^T O_2\Omega_{6[t-\delta_2,t-\delta(t)]}$$

$$+6\Omega_{7\,[t-\delta_2,t-\delta(t)]}^T O_2\Omega_{7[t-\delta_2,t-\delta(t)]} + 2\Omega_{8\,[t-\delta(t),t-\delta_1]}^T O_3\Omega_{8[t-\delta(t),t-\delta_1]}$$

$$+4\Omega_{9\,[t-\delta(t),t-\delta_1]}^T O_3\Omega_{9[t-\delta(t),t-\delta_1]} + 6\Omega_{10[t-\delta(t),t-\delta_1]}^T O_3\Omega_{10[t-\delta(t),t-\delta_1]}$$

$$+2\Omega_{8\,[t-\delta_2,t-\delta(t)]}^T O_3\Omega_{8[t-\delta_2,t-\delta(t)]} + 4\Omega_{9\,[t-\delta_2,t-\delta(t)]}^T O_3\Omega_{9[t-\delta_2,t-\delta(t)]}$$

$$+6\Omega_{10[t-\delta_2,t-\delta(t)]}^T O_3\Omega_{10[t-\delta_2,t-\delta(t)]}^T + \Omega_{1\,[t-\delta(t),t-\delta_1]}^T O_1\Omega_{1[t-\delta(t),t-\delta_1]}$$

$$+3\Omega_{2\,[t-\delta(t),t-\delta_1]}^T O_1\Omega_{2[t-\delta(t),t-\delta_1]} + 5\Omega_{3\,[t-\delta(t),t-\delta_1]}^T O_1\Omega_{3[t-\delta(t),t-\delta_1]}$$

$$+7\Omega_{4\,[t-\delta(t),t-\delta_1]}^T O_1\Omega_{4[t-\delta(t),t-\delta_1]} + \Omega_{1\,[t-\delta_2,t-\delta(t)]}^T O_1\Omega_{1[t-\delta_2,t-\delta(t)]}$$

$$+3\Omega_{2\,[t-\delta_2,t-\delta(t)]}^T O_1\Omega_{2[t-\delta_2,t-\delta(t)]} + 5\Omega_{3\,[t-\delta_2,t-\delta(t)]}^T O_1\Omega_{3[t-\delta_2,t-\delta(t)]}$$

$$+7\Omega_{4\,[t-\delta_2,t-\delta(t)]}^T O_1\Omega_{4[t-\delta_2,t-\delta(t)]} + \Omega_{1\,[t-\delta(t),t-\delta_1]}^T Z_1\Omega_{1[t-\delta_2,t-\delta(t)]}$$

$$+\Omega_{1\,[t-\delta_2,t-\delta(t)]}^T Z_1^T\Omega_{1[t-\delta(t),t-\delta_1]} + 3\Omega_{2\,[t-\delta(t),t-\delta_1]}^T Z_2\Omega_{2[t-\delta_2,t-\delta(t)]}$$

$$+3\Omega_2{}^T_{[t-\delta_2,t-\delta(t)]}Z_2^T\Omega_{2[t-\delta(t),t-\delta_1]} + 5\Omega_3{}^T_{[t-\delta(t),t-\delta_1]}Z_3\Omega_{3[t-\delta_2,t-\delta(t)]}$$

$$+5\Omega_3{}^T_{[t-\delta_2,t-\delta(t)]}Z_3^T\Omega_{3[t-\delta(t),t-\delta_1]} + 7\Omega_4{}^T_{[t-\delta(t),t-\delta_1]}Z_4\Omega_{4[t-\delta_2,t-\delta(t)]}$$

$$+7\Omega_4{}^T_{[t-\delta_2,t-\delta(t)]}Z_4^T\Omega_{4[t-\delta(t),t-\delta_1]}\Big\} - 2\alpha V_4(z_t).$$

The times derivative of $V_5(z_t)$ is calculated as

$$\dot{V}_5(z_t) \quad \leq \quad \dot{z}^T(t)(S_1 + S_2 + S_3)\dot{z}(t) - e^{-2\alpha\tau_2}\dot{z}^T(t-\tau_2)S_1\dot{z}(t-\tau_2)$$

$$-e^{-2\alpha\tau_2}\dot{z}^T(t-\tau(t))S_2\dot{z}(t-\tau(t)) + \tau_d\dot{z}^T(t-\tau(t))S_2\dot{z}(t-\tau(t))$$
$$-e^{-2\alpha\tau_1}\dot{z}^T(t-\tau_1)S_3\dot{z}(t-\tau_1) - 2\alpha V_5(z_t).$$

Further, from Lemma 1, we receive $\dot{V}_6(z_t)$ as follows

$$\dot{V}_6(z_t) \quad \leq \quad g^T(z(t))\Big[\eta_2^2 T_1 + \eta_2^2 T_2 + \eta_1^2 T_3\Big]g(z(t)) - e^{-2\alpha\eta_2}\int_{t-\eta_2}^t g^T(z(s))ds\,T_1$$

$$\times \int_{t-\eta_2}^t g(z(s))ds - e^{-2\alpha\eta_2}\int_{t-\eta(t)}^t g^T(z(s))ds\,T_2\int_{t-\eta(t)}^t g(z(s))ds$$

$$-e^{-2\alpha\eta_1}\int_{t-\eta_1}^t g^T(z(s))ds\,T_3\int_{t-\eta_1}^t g(z(s))ds - 2\alpha V_6(z_t).$$

Since $0 \leq \frac{f_i(s)}{s_i} \leq l_i$, $i = 1,2,\ldots,n$, for diagonal matrices $U_m > 0$, $m = 1,2,3,4$, we have

$$\Big[z^T(t)L - g^T(z(t))\Big]U_1^T g(z(t)) \geq 0, \tag{14}$$

$$\Big[z^T(t-\delta_1)L - g^T(z(t-\delta_1))\Big]U_2^T g(z(t-\delta_1)) \geq 0, \tag{15}$$

$$\Big[z^T(t-\delta(t))L - g^T(z(t-\delta(t)))\Big]U_3^T g(z(t-\delta(t))) \geq 0, \tag{16}$$

$$\Big[z^T(t-\delta_2)L - g^T(z(t-\delta_2))\Big]U_4^T g(z(t-\delta_2)) \geq 0. \tag{17}$$

Through the use zero equations, for any appropriate dimensions $X_k$, $k = 1,2,\ldots,7$, and $Y_l$, $l = i = 1,2,\ldots,5$, we obtain the following equations

$$2\Big[\dot{z}^T(t)X_1^T + z^T(t)X_2^T + g^T(z(t))X_3^T + g^T(z(t-\delta(t)))X_4^T + \dot{z}^T(t-\tau(t))X_5^T$$

$$+\int_{t-\eta(t)}^t g^T(z(s))ds\,X_6^T + u^T(t)X_7^T\Big]\Big[-\dot{z}(t) - Az(t) + W_0 g(z(t))$$

$$+W_1 g(z(t-\delta(t))) + W_2\dot{z}(t-\tau(t)) + W_3\int_{t-\eta(t)}^t g(z(s))ds + u(t)\Big] = 0, \tag{18}$$

$$2\Big[\dot{z}^T(t)Y_1^T + w^T(t)Y_2^T\Big]\Big[\dot{z}(t) - w(t)\Big] = 0, \tag{19}$$

$$2\Big[z^T(t)Y_3^T + z^T(t-\delta(t))Y_4^T + \int_{t-\delta(t)}^t \dot{z}^T(s)ds\,Y_5^T\Big]\Big[z(t) - z(t-\delta(t))$$

$$-\int_{t-\delta(t)}^t \dot{z}(s)ds\Big] = 0. \tag{20}$$

Recalling (14)–(20) and estimation of the time derivative of $V(z_t)$, it is apparent that

$$\dot{V}(z_t) + 2\alpha V(z_t) - 2y(t)u(t) \quad \leq \quad \xi^T(t)\Lambda\xi(t), \tag{21}$$

where $\xi(t) = \big[z(t), z(t - \delta_1), z(t - \delta(t)), z(t - \delta_2), g(z(t)), g(z(t - \delta_1)), g(z(t - \delta(t))),$

$g(z(t - \delta_2)), \dot{z}(t), \int_{t-\delta_2}^{t} \dot{z}(s)ds, \omega(t), \frac{1}{\delta_1} \int_{t-\delta_1}^{t} z(s)ds, \frac{1}{\delta_1^2} \int_{t-\delta_1}^{t} \int_{\theta}^{t} z(s)dsd\theta, \frac{1}{\delta_1^3} \int_{t-\delta_1}^{t} \int_{u}^{t} \int_{\theta}^{t} z(s)ds$

$d\theta du, \frac{1}{\delta(t)-\delta_1} \int_{t-\delta(t)}^{t-\delta_1} z(s)ds, \frac{1}{(\delta(t)-\delta_1)^2} \int_{t-\delta(t)}^{t-\delta_1} \int_{\theta}^{t-\delta_1} z(s)dsd\theta, \frac{1}{(\delta(t)-\delta_1)^3} \times \int_{t-\delta(t)}^{t-\delta_1} \int_{u}^{t-\delta_1} \int_{\theta}^{t-\delta_1} z(s)$

$dsd\theta du, \frac{1}{\delta_2-\delta(t)} \int_{t-\delta_2}^{t-\delta(t)} z(s)ds, \frac{1}{(\delta_2-\delta(t))^2} \int_{t-\delta_2}^{t-\delta(t)} \int_{\theta}^{t-\delta(t)} z(s)dsd\theta, \frac{1}{(\delta_2-\delta(t))^3} \int_{t-\delta_2}^{t-\delta(t)} \int_{u}^{t-\delta(t)}$

$\int_{\theta}^{t-\delta(t)} z(s)dsd\theta du, \dot{z}(t - \tau_1), \dot{z}(t - \tau(t)), \dot{z}(t - \tau_2), \int_{t-\eta_1}^{t} g(z(s))ds, \int_{t-\eta(t)}^{t} g(z(s))ds, \int_{t-\eta_2}^{t}$

$g(z(s))ds, u(t)\big]$ and $\Lambda$ is defined in (9). From assumption (21), it is readily visible that

$$\dot{V}(z_t) + 2\alpha V(z_t) - 2y(t)u(t) \leq 0, \qquad t \geq 0,$$

or

$$\dot{V}(z_t) + \rho V(z_t) \leq 2y(t)u(t), \qquad t \geq 0,$$

where $\rho = 2\alpha$. Therefore, if the LMIs conditions (9) and (10) hold, we conclude that the system (11) is exponentially passive. This proof is complete. □

Moreover, we suggest the robust exponential passivity for uncertain NTNNs with mixed interval time-varying delays of system (1). It is apparent that system (1) is robustly exponentially passive, from which we summarize the corresponding result in Theorem 2.

**Theorem 2.** *Assume that Assumptions 1 and 2 hold. For scalars $\delta_1, \delta_2, \delta_d, \tau_1, \tau_2, \tau_d, \eta_1, \eta_2, \rho,$ $\beta$ with conditions (2)–(4), $\rho > 0$ and $\beta > 0$, if there exist matrices $P > 0$, $M_n$, $n = 1, 2, \cdots, 6,$ $\begin{bmatrix} M_1 & M_2 \\ * & M_3 \end{bmatrix} > 0, \begin{bmatrix} M_4 & M_5 \\ * & M_6 \end{bmatrix} > 0, \ N_j > 0, O_j > 0, S_j > 0, T_j > 0, j = 1, 2, 3,$ any diagonal matrices $D_1 > 0, D_2 > 0, U_i > 0, i = 1, 2, 3, 4,$ any appropriate dimensional matrices $X_k, Y_l, Z_i,$ $k = 1, 2, \cdots, 7, l = 1, 2, \cdots, 5, i = 1, 2, 3, 4,$ satisfying LMIs (9), (10) and*

$$\begin{bmatrix} \Lambda & \Gamma_1 & \beta\Gamma_2^T \\ * & -\beta I & \beta J^T \\ * & * & -\beta I \end{bmatrix} < 0, \tag{22}$$

*then the system (1) is robust exponential passive.*

**Proof.** Based on Theorem 1, if $A, W_0, W_1, W_2,$ and $W_3$ are replaced with $A + E\Delta(t)G_a, W_0 + E\Delta(t)G_0, W_1 + E\Delta(t)G_1, W_2 + E\Delta(t)G_2$ and $W_3 + E\Delta(t)G_3,$ respectively, a new criterion of the uncertain NTNNs (1) is equivalent to the following conditions,

$$\Lambda + \Gamma_1 \Delta(t) \Gamma_2 + \Gamma_2^T \Delta(t)^T \Gamma_1^T \leq 0, \tag{23}$$

where $\Gamma_1 = \big[(P + Q_9 + X_2)^T E, 0, 0, 0, (Q_{11} + X_3)^T E, 0, (Q_{12} + X_4)^T E, 0, X_1^T E, Q_{10}^T E,$ $\underbrace{0, 0, \ldots, 0}_{11\,items}, X_5^T E, 0, 0, X_6^T E, 0, X_7^T E\big]$ and $\Gamma_2 = \big[ -G_a, 0, 0, 0, G_0, 0, G_1, \underbrace{0, 0, \ldots, 0}_{14\,items}, G_2, 0, 0,$ $G_3, 0, 0\big].$ By Lemma 4, an adequate condition ensuring

$$\begin{bmatrix} \Lambda & \Gamma_1 & \beta\Gamma_2^T \\ * & -\beta I & \beta J^T \\ * & * & -\beta I \end{bmatrix} < 0,$$

is that there exists a scalar $\beta > 0$. Together with the similar proof of Theorem 1, we have

$$\dot{V}(z_t) + \rho V(z_t) \leq 2y(t)u(t), \quad t \geq 0.$$

Based on Definition 1, the uncertain NTNNs (1) is robustly exponentially passive. This completes the proof. □

## 4. Numerical Examples

In this section, we allow the numerical examples to show the performance of the systems (1) and (11).

**Example 1.** *Consider the NTNNs (11), with the parameters [2,7,24] being as follows:*

$$A = \begin{bmatrix} 2 & 0 \\ 0 & 2 \end{bmatrix}, \ W_0 = \begin{bmatrix} \gamma & 0.3 \\ 0.3 & 0.5 \end{bmatrix}, \ W_1 = \begin{bmatrix} 0.2 & 0.1 \\ 0.1 & 0.2 \end{bmatrix}, \ W_2 = \begin{bmatrix} 0.15 & 0 \\ 0 & 0.15 \end{bmatrix},$$

$$W_3 = \begin{bmatrix} 0 & 0 \\ 0 & 0 \end{bmatrix}, \ L = \begin{bmatrix} 1 & 0 \\ 0 & 1 \end{bmatrix}, \ u(t) = 0, \ y(t) = 0.$$

*By applying Theorem 1, we find that the LMIs (9) and (10) are feasible. In Table 1, we compare the maximum allowable bound $\gamma$, where $\delta_1 = 0.5$, $\delta_2 = 2.0$, $\tau_1 = 0.5$, $\tau_2 = 1.0$, and $\delta_d = 0.9$ for ensuring Theorem 1 of the system (11), which the potential of our result exceeds those previous results ([2,7,24]).*

**Table 1.** The maximum upper bound of $\gamma$ for different $\tau_d$.

| $\tau_d$ | 0.5 | 0.8 |
|---|---|---|
| Park and Kwon [2] | 1.65 | - |
| Tu et al. [24] | 2.66 | - |
| Manivannan et al. [7] | 3.94 | 3.43 |
| Theorem 1 | 4.06 | 3.68 |

**Example 2.** *Considering the uncertain NTNNs (1), the parameters [22] are as follows:*

$$A = \begin{bmatrix} 4 & 0 \\ 0 & 4 \end{bmatrix}, \ W_0 = \begin{bmatrix} 0.6 & 0.4 \\ -0.5 & 0.4 \end{bmatrix}, \ W_1 = \begin{bmatrix} 0.9993 & 0.3554 \\ 0.0471 & -0.2137 \end{bmatrix}, \ W_2 = \begin{bmatrix} 0 & 0 \\ 0 & 0 \end{bmatrix},$$

$$W_3 = \begin{bmatrix} 0.3978 & 1.3337 \\ -0.2296 & 0.9361 \end{bmatrix}, \ G_a = \begin{bmatrix} 0.2 & 0 \\ 0 & 0.2 \end{bmatrix}, \ G_0 = \begin{bmatrix} 0.6 & 0 \\ 0 & 0.6 \end{bmatrix}, \ G_1 = \begin{bmatrix} 0.3 & 0 \\ 0 & 0.3 \end{bmatrix},$$

$$G_2 = \begin{bmatrix} 0 & 0 \\ 0 & 0 \end{bmatrix}, \ G_3 = \begin{bmatrix} 0.4 & 0 \\ 0 & 0.4 \end{bmatrix}, \ L = E = C_0 = C_1 = C_3 = I, \ C_2 = 0.$$

*Through adaption of Theorem 2 and using the Matlab LMI toolbox, the feasibility of the aimed method holds. A similar result of this method was shown previously in [22] where $\delta(t) = 1 + \sin(t)$, and $\eta(t) = 1.5 + \cos(t)$. Meanwhile, we show that the robust exponential passive of system (1) is guaranteed by calculating the maximum allowable bound of $\rho$ for $\delta_2 = 2.0$, $\eta_1 = 0.0$, $\eta_2 = 2.5$, different $\delta_d$, and various $\delta_1$ in Table 2.*

**Table 2.** The maximum upper bound of $\rho$ for different $\delta_d$ and various $\delta_1$.

| $\delta_d$ | $\delta_1 = 0.0$ | $\delta_1 = 0.1$ | $\delta_1 = 0.2$ |
|---|---|---|---|
| 0.1 | 0.5224 | 0.9252 | 0.4510 |
| 0.3 | 0.3204 | 0.5790 | 0.3782 |
| 0.5 | 0.1734 | 0.3362 | 0.2642 |
| 0.8 | 0.0092 | 0.1080 | 0.0804 |

**Example 3.** *Consider the uncertain NTNNs (1), with the parameters [21,22] being as follows:*

$$A = \begin{bmatrix} 4 & 0 \\ 0 & 7 \end{bmatrix}, \ W_0 = \begin{bmatrix} 0 & -0.5 \\ 0.5 & 0 \end{bmatrix}, \ W_1 = \begin{bmatrix} -1 & -1 \\ -1 & -2 \end{bmatrix}, \ W_2 = W_3 = \begin{bmatrix} 0 & 0 \\ 0 & 0 \end{bmatrix},$$

$$G_a = \begin{bmatrix} 0.02 & 0.04 \\ 0.03 & 0.06 \end{bmatrix}, \ G_0 = \begin{bmatrix} 0.02 & 0.04 \\ 0.02 & 0.04 \end{bmatrix}, \ G_1 = \begin{bmatrix} 0.03 & 0.06 \\ 0.02 & 0.04 \end{bmatrix}, \ G_2 = G_3 = 0,$$

$$L = E = I, \ C_0 = C_1 = C_3 = I, \ C_2 = 0.$$

*This example shows the maximum bounds of the allowable values $\rho$ where $\delta_1 = 0.0$ and $\delta_2 = 0.16$ to guarantee that system (1) is the robust exponential passive. The list in Table 3 shows that the potential of our results is superior to those in [21,22].*

**Table 3.** The maximum upper bound of $\rho$ for different $\delta_d$.

| $\delta_d$ | 0.1 | 0.3 | 0.5 | 0.9 |
|---|---|---|---|---|
| Wu et al. [21] | 5.4753 | 5.4121 | 5.3518 | 5.2864 |
| Du et al. [22] ($N = 3, M = 3$) | 6.0297 | 5.8124 | 5.7735 | 5.7294 |
| Theorem 2 | 6.0374 | 5.9652 | 5.9634 | 5.9634 |

**Example 4.** *Consider the uncertain NTNNs (1), the parameters being as follows:*

$$A = \begin{bmatrix} 4 & 0 \\ 0 & 5 \end{bmatrix}, \; W_0 = \begin{bmatrix} -0.4 & 0 \\ -0.1 & 0.1 \end{bmatrix}, \; W_1 = \begin{bmatrix} 0.1 & 0.2 \\ -0.15 & -0.18 \end{bmatrix}, \; W_2 = \begin{bmatrix} 0.1 & 0 \\ 0 & 0.1 \end{bmatrix},$$

$$W_3 = \begin{bmatrix} 0.41 & -0.5 \\ 0.69 & 0.31 \end{bmatrix}, \; G_a = \begin{bmatrix} 0 & 0 \\ 0.1 & 0.1 \end{bmatrix}, \; G_0 = G_1 \begin{bmatrix} 0 & 0 \\ 0.02 & 0.03 \end{bmatrix},$$

$$G_2 = \begin{bmatrix} 0 & 0 \\ 0.001 & 0.001 \end{bmatrix}, \; G_3 = \begin{bmatrix} 0 & 0 \\ 0.02 & 0.02 \end{bmatrix}, \; L = E = C_0 = C_1 = C_2 = C_3 = I.$$

*We confirm the feasibility of the criterion of Theorem 2 by using the Matlab LMI toolbox, where $0.5 \leq \delta(t) \leq 4$, $\delta_d = 0.5$, $0.5 \leq \tau(t) \leq 3.5$, $\tau_d = 0.7$, $0.5 \leq \eta(t) \leq 1$, $\rho = 0.094$, of which the feasible solution is as follows:*

$$P = \begin{bmatrix} 1.3322 & 0.2119 \\ 0.2119 & 1.0924 \end{bmatrix}, \qquad D_1 = \begin{bmatrix} 1.4453 & 0 \\ 0 & 1.4453 \end{bmatrix},$$

$$D_2 = \begin{bmatrix} 0.9707 & 0 \\ 0 & 0.9707 \end{bmatrix}, \qquad M_1 = \begin{bmatrix} 0.1220 & 0.0042 \\ 0.0042 & 0.1062 \end{bmatrix},$$

$$M_2 = \begin{bmatrix} -0.1326 & 0.0016 \\ 0.0016 & -0.1384 \end{bmatrix}, \qquad M_3 = \begin{bmatrix} 0.3416 & -0.0051 \\ -0.0051 & 0.3610 \end{bmatrix},$$

$$M_4 = \begin{bmatrix} 4.7528 & 0.3159 \\ 0.3159 & 4.4572 \end{bmatrix}, \qquad M_5 = \begin{bmatrix} -2.0930 & -0.2243 \\ -0.2243 & -0.9815 \end{bmatrix},$$

$$M_6 = \begin{bmatrix} 8.2592 & 0.9770 \\ 0.9770 & 7.3324 \end{bmatrix}, \qquad N_1 = \begin{bmatrix} 0.0509 & 0.0031 \\ 0.0031 & 0.0370 \end{bmatrix},$$

$$N_2 = 10^{-3} \begin{bmatrix} 0.7335 & 0.0602 \\ 0.0602 & 0.5430 \end{bmatrix}, \qquad N_3 = \begin{bmatrix} 0.0769 & 0.0033 \\ 0.0033 & 0.0656 \end{bmatrix},$$

$$O_1 = \begin{bmatrix} 0.0012 & 0.0001 \\ 0.0001 & 0.0008 \end{bmatrix}, \qquad O_2 = \begin{bmatrix} 0.0016 & 0.0001 \\ 0.0001 & 0.0011 \end{bmatrix},$$

$$O_3 = \begin{bmatrix} 0.0016 & 0.0001 \\ 0.0001 & 0.0011 \end{bmatrix}, \qquad S_1 = \begin{bmatrix} 0.4670 & 0.0273 \\ 0.0273 & 0.3901 \end{bmatrix},$$

$$S_2 = \begin{bmatrix} 0.0040 & 0.0003 \\ 0.0003 & 0.0028 \end{bmatrix}, \qquad S_3 = \begin{bmatrix} 0.0039 & 0.0003 \\ 0.0003 & 0.0027 \end{bmatrix},$$

$$T_1 = \begin{bmatrix} 0.2694 & 0.0200 \\ 0.0200 & 0.1852 \end{bmatrix}, \qquad T_2 = \begin{bmatrix} 3.2174 & 0.0233 \\ 0.0233 & 2.5143 \end{bmatrix},$$

$$T_3 = \begin{bmatrix} 1.0763 & 0.0798 \\ 0.0798 & 0.7395 \end{bmatrix}, \qquad Q_1 = 10^3 \begin{bmatrix} -1.1925 & 0.0000 \\ 0.0000 & -1.1924 \end{bmatrix},$$

$$Q_2 = 10^3 \begin{bmatrix} 1.1927 & 0.0001 \\ 0.0001 & 1.1926 \end{bmatrix}, \qquad Q_3 = \begin{bmatrix} -0.5234 & -0.0422 \\ -0.0422 & -0.4805 \end{bmatrix},$$

$$Q_4 = \begin{bmatrix} 0.2096 & 0.0098 \\ 0.0098 & 0.0691 \end{bmatrix}, \qquad Q_5 = 10^3 \begin{bmatrix} -1.1923 & 0.0000 \\ 0.0000 & -1.1924 \end{bmatrix},$$

$$Q_6 = 10^3 \begin{bmatrix} 1.1927 & -0.0000 \\ -0.0000 & 1.1928 \end{bmatrix}', \qquad Q_7 = \begin{bmatrix} -0.5771 & -0.0465 \\ -0.0465 & -0.3998 \end{bmatrix}',$$

$$Q_8 = \begin{bmatrix} 0.2180 & 0.0171 \\ 0.0171 & 0.0950 \end{bmatrix}', \qquad Q_9 = \begin{bmatrix} 0.9438 & -0.6339 \\ -0.6339 & -0.7053 \end{bmatrix}',$$

$$Q_{10} = \begin{bmatrix} -0.0146 & -0.0911 \\ -0.0911 & 0.0433 \end{bmatrix}', \qquad Q_{11} = \begin{bmatrix} -0.4912 & 0.8620 \\ 0.8620 & 0.6054 \end{bmatrix}',$$

$$Q_{12} = \begin{bmatrix} -0.7129 & -0.3630 \\ -0.3630 & -1.0636 \end{bmatrix}', \qquad Z_1 = 10^{-3} \begin{bmatrix} -0.1163 & -0.0042 \\ -0.0042 & -0.0458 \end{bmatrix}',$$

$$Z_2 = 10^{-6} \begin{bmatrix} -0.2203 & 0.0787 \\ 0.0787 & -0.5239 \end{bmatrix}', \qquad Z_3 = 10^{-8} \begin{bmatrix} -0.0204 & 0.0252 \\ 0.0252 & -0.1607 \end{bmatrix}',$$

$$Z_4 = 10^{-9} \begin{bmatrix} -0.4913 & -0.0730 \\ -0.0730 & -0.2132 \end{bmatrix}', \qquad U_1 = \begin{bmatrix} 8.1698 & 0 \\ 0 & 8.1698 \end{bmatrix}',$$

$$U_2 = \begin{bmatrix} 0.1671 & 0 \\ 0 & 0.1671 \end{bmatrix}', \qquad U_3 = \begin{bmatrix} 0.5990 & 0 \\ 0 & 0.5990 \end{bmatrix}',$$

$$U_4 = \begin{bmatrix} 0.0479 & 0 \\ 0 & 0.0479 \end{bmatrix}', \qquad X_1 = \begin{bmatrix} 6.4830 & -0.7995 \\ -0.7995 & 4.5869 \end{bmatrix}',$$

$$X_2 = \begin{bmatrix} 25.5529 & -4.0582 \\ -4.0582 & 21.7320 \end{bmatrix}', \qquad X_3 = \begin{bmatrix} 2.0164 & 0.5346 \\ 0.5346 & -0.2008 \end{bmatrix}',$$

$$X_4 = \begin{bmatrix} -1.5646 & -0.1081 \\ -0.1081 & 0.2932 \end{bmatrix}', \qquad X_5 = \begin{bmatrix} -0.6444 & 0.0799 \\ 0.0799 & -0.4607 \end{bmatrix}',$$

$$X_6 = \begin{bmatrix} -2.9287 & 0.5075 \\ 0.5075 & -1.7624 \end{bmatrix}', \qquad X_7 = \begin{bmatrix} -6.6621 & 0.9600 \\ 0.9600 & -4.4849 \end{bmatrix}',$$

$$Y_1 = 10^3 \begin{bmatrix} -5.3746 & -0.0050 \\ -0.0050 & -5.3748 \end{bmatrix}', \qquad Y_2 = 10^3 \begin{bmatrix} 5.3750 & 0.0050 \\ 0.0050 & 5.3750 \end{bmatrix}',$$

$$Y_3 = 10^3 \begin{bmatrix} -1.1923 & 0.0000 \\ 0.0000 & -1.1924 \end{bmatrix}', \qquad Y_4 = 10^3 \begin{bmatrix} 3.5779 & 0.0000 \\ 0.0000 & 3.5778 \end{bmatrix}',$$

$$Y_5 = 10^3 \begin{bmatrix} 1.1927 & -0.0000 \\ -0.0000 & 1.1927 \end{bmatrix}', \qquad \text{and } \beta = 87.2598.$$

Figure 1 gives the state response of the uncertain NTNNs (1) under zero input, and the initial condition $[-0.2, 0.2]$, the interval time-varying delays are chosen as $\delta(t) = 3 + 0.25|\sin(2t)|$, $\tau(t) = 3 + 0.35|\sin(2t)|$, and $\eta(t) = |\cos(t)|$, the activation function is set as $g(z(t)) = \tanh(z_(t))$. Meanwhile, we sketched the solution trajectory of the system (1) of Example 4 with initial conditions $[3.5, 3.5]$ in Figure 2.

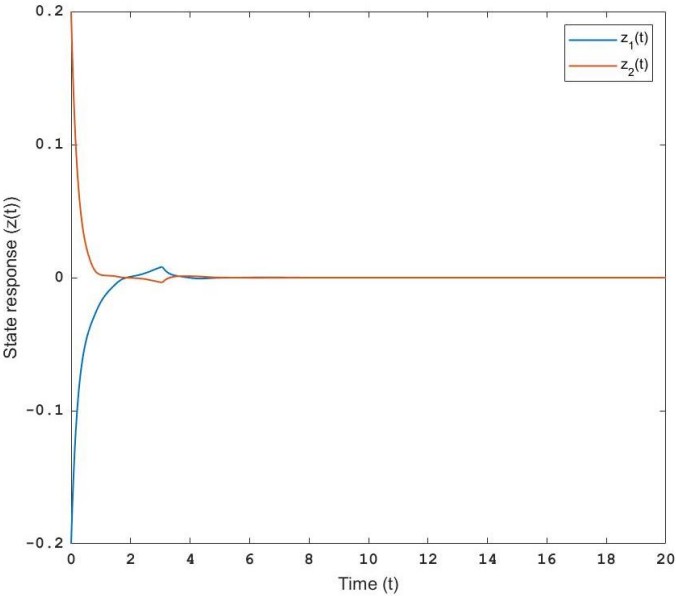

**Figure 1.** The state response of the system (1).

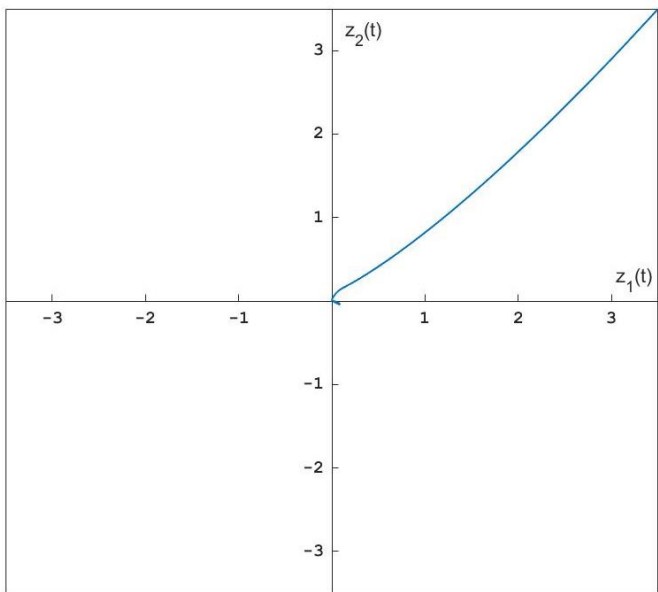

**Figure 2.** The solution trajectory of the system (1).

## 5. Conclusions

In this paper, we considered the robust exponential passivity analysis for uncertain NTNNs with mixed interval time-varying delays including discrete, neutral, and distributed delays. We concentrated on interval time-varying delays, in which the lower bounds were allowed to be either positive or zero. For the potential of results, we simultaneously adopted the model transformation, various inequalities, the reciprocally convex combination, and suitable Lyapunov-–Krasovskii functional. In the first place, a new exponential passivity analysis for NTNNs was derived and formulated in the form of LMIs. Secondly, a new robust exponential passivity analysis for uncertain NTNNs was obtained. Note that our proposed method can be adapted to many criteria such as exponential passivity of uncertain neural networks with distributed time-varying delays, the exponential passivity of uncertain neural networks with time-varying delays, stability of NTNNs with interval time-varying delays. Lastly, the feasibility of the aimed methods was shown in numerical simulations by applying our method. We achieved the aim to receive the maximum values of the rate of convergence, for which our results exceeded the results that were previously seen. We also introduced a new example to demonstrate the existence of a solution to the proposed method.

**Author Contributions:** Conceptualization, N.S.; methodology, N.S.; software, N.S. and N.Y.; validation, N.S. and K.M.; formal analysis, K.M., ; investigation, N.S., N.Y., P.S. and K.M.; writing—original draft preparation, N.S., N.Y., P.S. and K.M.; writing—review and editing, N.S., N.Y., P.S. and K.M.; visualization, N.Y., P.S., and K.M.; supervision, N.Y. and K.M.; project administration, N.S., N.Y., P.S. and K.M.; funding acquisition, K.M. All authors have read and agreed to the published version of the manuscript.

**Funding:** This work is supported by National Research Council of Thailand (NRCT) and Khon Kaen University (Mid-Career Research Grant NRCT5-RSA63003).

**Institutional Review Board Statement:** Not applicable.

**Informed Consent Statement:** Not applicable.

**Data Availability Statement:** Not applicable.

**Acknowledgments:** The authors thank the reviewers for their valuable comments and suggestions, which led to the improvement of the content of the paper.

**Conflicts of Interest:** The authors declare no conflict of interest.

## Appendix A

$$\Lambda = [\Lambda_{i,j}]_{27\times27}, \quad \Lambda_{i,j} = \Lambda_{j,i}^T, \quad i,j = 1,2,\ldots,27,$$

$$\Lambda_{1,1} = 2\alpha P - (P + Q_9 + X_2)^T A - A^T(P + Q_9 + X_2) + Q_1 + Q_1^T + Q_5 + Q_5^T$$
$$+ M_4 + 2\alpha L^T D_2^T + 2\alpha D_2 L + Y_3 + Y_3^T - e^{-2\alpha\delta_1}(16N_1 + 12N_2),$$

$$\Lambda_{1,2} = -4e^{-2\alpha\delta_1}N_1, \quad \Lambda_{1,3} = -Q_1^T - Q_5^T - Y_3^T + Y_4, \quad \Lambda_{1,5} = (P + Q_9 + X_2)^T W_0$$
$$+ Q_3 + Q_7 - A^T(Q_{11} + X_3) + 2\alpha D_1^T - 2\alpha D_2 + M_5 + LU_1^T,$$

$$\Lambda_{1,7} = (P + Q_9 + X_2)^T W_1 + Q_4 + Q_8 - A^T(Q_{12} + X_4), \quad \Lambda_{1,9} = D_2 L - X_2^T$$
$$- A^T X_1, \quad \Lambda_{1,10} = -Q_1^T - Q_5^T + Q_2 + Q_6 - AQ_{10} - Y_3^T + Y_5, \quad \Lambda_{1,11} = -Q_9,$$

$$\Lambda_{1,12} = e^{-2\alpha\delta_1}(60N_1 + 12N_2), \quad \Lambda_{1,13} = -e^{-2\alpha\delta_1}(360N_1 + 120N_2),$$

$$\Lambda_{1,14} = e^{-2\alpha\delta_1}(840N_1 + 360N_2), \quad \Lambda_{1,22} = (P + Q_9 + X_2)^T W_2 - A^T X_5,$$

$$\Lambda_{1,25} = (P + Q_9 + X_2)^T W_3 - A^T X_6, \quad \Lambda_{1,27} = P^T + Q_9^T + X_2^T - A^T X_7,$$

$$\Lambda_{2,2} = e^{-2\alpha\delta_1}M_1 - e^{-2\alpha\delta_1}(16N_1 + 12N_3) - e^{-2\alpha\delta_2}(16O_1 + 12O_2),$$

$$\Lambda_{2,3} = -e^{-2\alpha\delta_2}(Z_1 + 3Z_2 + 5Z_3 + 7Z_4)^T - 4e^{-2\alpha\delta_2}O_1, \quad \Lambda_{2,4} = e^{-2\alpha\delta_2}(Z_1 - 3Z_2$$
$$+ 5Z_3 - 7Z_4)^T, \quad \Lambda_{2,6} = e^{-2\alpha\delta_1}M_2 + LU_2^T, \quad \Lambda_{2,12} = e^{-2\alpha\delta_1}(120N_1 + 72N_3),$$

$$\Lambda_{2,13} = -e^{-2\alpha\delta_1}(480N_1 + 240N_3), \quad \Lambda_{2,14} = e^{-2\alpha\delta_1}(840N_1 + 360N_3),$$

$$\Lambda_{2,15} = e^{-2\alpha\delta_2}(60O_1 + 12O_2), \quad \Lambda_{2,16} = -e^{-2\alpha\delta_2}(360O_1 + 120O_2),$$

$$\Lambda_{2,17} = e^{-2\alpha\delta_2}(840O_1 + 360O_2), \quad \Lambda_{2,18} = e^{-2\alpha\delta_2}(6Z_2 - 30Z_3 + 84Z_4)^T,$$

$$\Lambda_{2,19} = e^{-2\alpha\delta_2}(60Z_3 - 420Z_4)^T, \quad \Lambda_{2,20} = 840e^{-2\alpha\delta_2}Z_4^T, \quad \Lambda_{3,3} = (\delta_d - e^{-2\alpha\delta_2})M_4$$
$$- Y_4 - Y_4^T + e^{-2\alpha\delta_2}(Z_1 - 3Z_2 + 5Z_3 - 7Z_4)^T + e^{-2\alpha\delta_2}(Z_1 - 3Z_2 + 5Z_3$$
$$- 7Z_4) - e^{-2\alpha\delta_2}(32O_1 + 12O_2 + 12O_3), \quad \Lambda_{3,4} = -e^{-2\alpha\delta_2}(Z_1 + 3Z_2 + 5Z_3$$
$$+ 7Z_4)^T - 4e^{-2\alpha\delta_2}O_1, \quad \Lambda_{3,5} = -Q_3 - Q_7, \quad \Lambda_{3,7} = -Q_4 - Q_8 + (\delta_d$$
$$- e^{-2\alpha\delta_2})M_5 + LU_3^T, \quad \Lambda_{3,10} = -Q_2 - Q_6 - Y_4^T - Y_5, \quad \Lambda_{3,15} = e^{-2\alpha\delta_2}(6Z_2$$
$$- 30Z_3 + 84Z_4)^T + e^{-2\alpha\delta_2}(120O_1 + 72O_3), \quad \Lambda_{3,16} = e^{-2\alpha\delta_2}(60Z_3 - 420Z_4)$$
$$- e^{-2\alpha\delta_2}(480O_1 + 240O_3), \quad \Lambda_{3,17} = 840e^{-2\alpha\delta_2}Z_4 + e^{-2\alpha\delta_2}(840O_1 + 360O_3),$$

$$\Lambda_{3,18} = e^{-2\alpha\delta_2}(6Z_2 + 30Z_3 + 84Z_4)^T + e^{-2\alpha\delta_2}(60O_1 + 12O_2),$$

$$\Lambda_{3,19} = -e^{-2\alpha\delta_2}(60Z_3 + 420Z_4)^T - e^{-2\alpha\delta_2}(360O_1 + 120O_2),$$

$$\Lambda_{3,20} = 840e^{-2\alpha\delta_2}Z_4^T + e^{-2\alpha\delta_2}(840O_1 + 360O_2), \quad \Lambda_{4,4} = -e^{-2\alpha\delta_2}M_1$$
$$- e^{-2\alpha\delta_2}(16O_1 + 12O_3), \quad \Lambda_{4,8} = -e^{-2\alpha\delta_2}M_2 + LU_4^T, \quad \Lambda_{4,15} = e^{-2\alpha\delta_2}(6Z_2$$
$$+ 30Z_3 + 84Z_4), \quad \Lambda_{4,16} = -e^{-2\alpha\delta_2}(60Z_3 + 420Z_4), \quad \Lambda_{4,17} = 840e^{-2\alpha\delta_2}Z_4,$$

$$\Lambda_{4,18} = e^{-2\alpha\delta_2}(120O_1 + 72O_3), \quad \Lambda_{4,19} = -e^{-2\alpha\delta_2}(480O_1 + 240O_3),$$

$$\Lambda_{4,20} = e^{-2\alpha\delta_2}(840O_1 + 360O_3), \quad \Lambda_{5,5} = (Q_{11} + X_3)^T W_0 + W_0^T(Q_{11} + X_3)$$
$$+ M_6 - U_1 - U_1^T + \eta_2^2(T_1 + T_2) + \eta_1^2 T_3, \quad \Lambda_{5,7} = (Q_{11}^T + X_3)^T W_1$$
$$+ W_0^T(Q_{12} + X_4), \quad \Lambda_{5,9} = D_1 - D_2 - X_3^T + W_0^T X_1, \quad \Lambda_{5,10} = -Q_3^T - Q_7^T$$
$$+ W_0^T Q_{10}, \quad \Lambda_{5,11} = -Q_{11}^T, \quad \Lambda_{5,22} = (Q_{11} + X_3)^T W_2 + W_0^T X_5,$$

$$\Lambda_{5,25} = (Q_{11} + X_3)^T W_3 + W_0^T X_6, \quad \Lambda_{5,27} = W_0^T X_7 + X_3^T + Q_{11}^T - C_0^T,$$

$$\Lambda_{6,6} = e^{-2\alpha\delta_2}M_3 - U_2 - U_2^T, \quad \Lambda_{7,7} = (Q_{12} + X_4)^T W_1 + W_1^T(Q_{12} + X_4)$$
$$+ (\delta_d - e^{-2\alpha\delta_2})M_6 - U_3 - U_3^T, \quad \Lambda_{7,9} = -X_4^T + W_1^T X_1, \quad \Lambda_{7,10} = -Q_4 - Q_8$$
$$+ W_1^T Q_{10}, \quad \Lambda_{7,11} = -Q_{12}, \quad \Lambda_{7,22} = (Q_{12} + X_4)^T W_2 + W_1^T X_5,$$

$$\Lambda_{7,25} = (Q_{12} + X_4)^T W_3 + W_1^T X_6, \quad \Lambda_{7,27} = W_1^T X_7 + X_4^T + Q_{12}^T - C_1^T,$$

$$\Lambda_{8,8} = -e^{-2\alpha\delta_2}M_3^T - U_4 - U_4^T, \quad \Lambda_{9,9} = Y_1 + Y_1^T - X_1 - X_1^T + \delta_1^2 N_1 + \frac{\delta_1^2}{2}(N_2$$
$$+ N_3), \quad \Lambda_{9,11} = -Y_1^T + Y_2^T, \quad \Lambda_{9,22} = X_1^T W_2 - X_5, \quad \Lambda_{9,25} = X_1^T W_3 - X_6,$$

$$\Lambda_{9,27} = -X_7 + X_1, \quad \Lambda_{10,10} = -Q_2 - Q_2^T - Q_6 - Q_6^T - Y_5 - Y_5^T, \quad \Lambda_{10,11} = -Q_{10}^T,$$

$$\Lambda_{10,22} = Q_{10}^T W_2, \ \Lambda_{10,25} = Q_{10}^T W_3, \ \Lambda_{10,27} = Q_{10}, \ \Lambda_{11,11} = -Y_2 - Y_2^T + (\delta_2 - \delta_1)^2$$
$$\times O_1 + \frac{(\delta_2 - \delta_1)^2}{2}(O_2 + O_3) + S_1 + S_2 + S_3,$$
$$\Lambda_{12,12} = -e^{-2\alpha\delta_1}(1200N_1 + 720N_2 + 552N_3), \ \Lambda_{12,13} = e^{-2\alpha\delta_1}(5400N_1 + 480N_2$$
$$+2040N_3), \ \Lambda_{12,14} = -e^{-2\alpha\delta_1}(10080N_1 + 10080N_2 + 3240N_3),$$
$$\Lambda_{13,13} = -e^{-2\alpha\delta_1}(25920N_1 + 3600N_2 + 7920N_3), \ \Lambda_{13,14} = e^{-2\alpha\delta_1}(50400N_1$$
$$+8640N_2 + 12960N_3, \ \Lambda_{14,14} = -e^{-2\alpha\delta_1}(100800N_1 + 21600N_2 + 21600N_3),$$
$$\Lambda_{15,15} = -e^{-2\alpha\delta_2}(1200O_1 + 72O_2 + 552O_3), \ \Lambda_{15,16} = e^{-2\alpha\delta_2}(5400O_1 + 480O_2$$
$$+2040O_3), \ \Lambda_{15,17} = -e^{-2\alpha\delta_2}(10080O_1 + 1080O_2 + 3240O_3),$$
$$\Lambda_{15,18} = -e^{-2\alpha\delta_2}(12Z_2 + 180Z_3 + 1008Z_4)^T, \ \Lambda_{15,19} = e^{-2\alpha\delta_2}(360Z_3 + 5040Z_4)^T,$$
$$\Lambda_{15,20} = -10080e^{-2\alpha\delta_2}Z_4^T, \ \Lambda_{16,16} = -e^{-2\alpha\delta_2}(25920O_1 + 3600O_2 + 7920O_3),$$
$$\Lambda_{16,17} = e^{-2\alpha\delta_2}(50400O_1 + 8640O_2 + 12960O_3), \ \Lambda_{16,18} = e^{-2\alpha\delta_2}(360Z_3$$
$$+5040Z_4)^T, \ \Lambda_{16,19} = -e^{-2\alpha\delta_2}(720Z_3 + 25200Z_4)^T,$$
$$\Lambda_{16,20} = 50400e^{-2\alpha\delta_2}Z_4^T, \ \Lambda_{17,17} = -e^{-2\alpha\delta_2}(100800O_1 + 21600O_2 + 21600O_3),$$
$$\Lambda_{17,18} = -10080e^{-2\alpha\delta_2}Z_4^T, \ \Lambda_{17,19} = 50400e^{-2\alpha\delta_2}Z_4^T, \ \Lambda_{17,20} = -100800e^{-2\alpha\delta_2}Z_4^T,$$
$$\Lambda_{18,18} = -e^{-2\alpha\delta_2}(1200O_1 + 72O_2 + 552O_3), \ \Lambda_{18,19} = e^{-2\alpha\delta_2}(5400O_1 + 480O_2$$
$$+2040O_3), \ \Lambda_{18,20} = -e^{-2\alpha\delta_2}(10080O_1 + 1080O_2 + 3240O_3),$$
$$\Lambda_{19,19} = -e^{-2\alpha\delta_2}(25920O_1 + 3600O_2 + 7920O_3), \ \Lambda_{19,20} = e^{-2\alpha\delta_2}(50400O_1$$
$$+8640O_2 + 12960O_3), \ \Lambda_{20,20} = -e^{-2\alpha\delta_2}(100800O_1 + 21600O_2$$
$$+21600O_3), \ \Lambda_{21,21} = -e^{-2\alpha\tau_1}S_3, \ \Lambda_{22,22} = (\tau_d - e^{-2\alpha\tau_2})S_2 + X_5^T W_2$$
$$+W_2 X_5, \ \Lambda_{22,25} = X_5^T W_3 + W_2 X_6, \ \Lambda_{22,27} = X_5^T + W_2^T X_7,$$
$$\Lambda_{23,23} = -e^{-2\alpha\tau_2}S_1, \ \Lambda_{24,24} = -e^{-2\alpha\delta_2}T_3, \ \Lambda_{25,25} = -e^{-2\alpha\delta_2}T_2 + X_6^T W_3 + W_3^T X_6,$$
$$\Lambda_{25,27} = X_6^T + W_3^T X_7 - C_2^T, \ \Lambda_{26,26} = -e^{-2\alpha\delta_2}T_1, \ \Lambda_{27,27} = X_7 + X_7^T - C_3 - C_3^T,$$

and the other terms are 0.

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
