# Peer review of "LMI-Based Results on Robust Exponential Passivity of Uncertain Neutral-Type Neural Networks with Mixed Interval Time-Varying Delays via the Reciprocally Convex Combination Technique"

_computation, doi:10.3390/computation9060070_

Round 1

Reviewer 1 Report

The paper studies a robust model of the delayed neural network. The robust passivity is investigated.  The main result is based on Lyapunov functionals and presented with the help of LMIs. The work should be revised significantly

Comments:

  1. Evidence of the existence of the solutions of the system (1)
  2. line 79: The derivative of V in the definition is omitted
  3. in line 79 and throughout the paper. Please clarify the notion of the total derivative of functional $V$. Is it the right-upper derivative?
  4.  Lemma 1. Reformulate it. Jensen's inequality doesn't include a derivative of $z$. 
  5. line 89 "where any positive real constant $\beta$". Reformulate it as  follows "where $\beta$ is any positive real constant"
  6. To better grasp the main results, the statement and the proof of Theorem 1 should be structured. Namely, include in the statement of Theorem 1 the general form of the LMIs.  The entire form of the LMIs and the proof should be in Appendix.
  7. line 100: error in referencing. Is it system (11)?
  8. Section "Numerical Examples" should be improved. Present phase plots of the solutions. 
  9. Demonstrate please the notion of robust exponential passivity graphically. What does it mean in practice?
  10. Please clarify the relation between exponential robust stability and passivity supporting it graphically.
  11. What is the effect of the delay? Did you observe periodic and chaotic solutions? Some explanation is needed.

Reviewer 2 Report

This paper considers the robust exponential passivity analysis for uncertain NTNNs with mixed interval time-varying delays include discrete, neutral, and distributed delays. It is an interesting read. The results are useful but the presentation can be improved. I have some comments as below.

1. How is neutral type neural network particularly related to the current stutdy? 
2. The novelty and contribution of the work should be clearly highlighted in the introduction. In the current version, it is only very briefly touched.
3. As delayed dynamical system is the key point of the work. It should be noted there there are two types of time delays: information processing delay and information communication delay. This is studied in the paper On the delayed scaled consensus problems. What is the type of delay considered here? Would you expect similar results for the other type of delay? Suggest a comment.
4. In system (1), both the derivative of z(t) and z(t) are depicted. There is something wrong. Please clarify if wrong notation is used.
5. Is the first condition in (5) the same as Lipschitz condition?
6. In line 85, should there be a comma between the two conditions in the brackets?
7. In the statement of Theorem 1, there are too many Lambda elements in the matrix which makes the equation not readable. Try to give some explanations to each term or present them in a more readable manner.
8. Another comment related to the above is that there are many large coefficients in the equations. It is not clear whether they are optimal or not.
9. The fifth line below line 117 seems to be not correct. A term is missing. I may be wrong but please double check.
10. In addition to the calculation, I suggest providing numerical simulation results in terms of figures if possible. 

Round 2

Reviewer 1 Report

Dear authors,

The revised version of the paper is much better. Almost all my comments are taken into account. The most important are two comments:

  1. Numerical simulations should be combined with the corresponding phase plots. For example, example 4. It is better to present dependence $z_1$ vs $z_2$ on a plane (not $z_1$ vs $t$ and $z_2$ vs $t$)
  2. Conclusions should be improved. Your conclusions are like the abstract of the paper. Here you should to discuss and analyze the advantages and disadvantages, potential applications of your results

Reviewer 2 Report

The paper can be accepted. I have no further comment.

Author Response

The authors thank the reviewers for their valuable comments and suggestions, which led to the improvement of the content of the paper.